# Exploring the Efficacy of Pembrolizumab in Combination with Carboplatin and Weekly Paclitaxel for Frail Patients with Advanced Non-Small-Cell Lung Cancer: A Key Investigative Study

**DOI:** 10.3390/cancers16050992

**Published:** 2024-02-29

**Authors:** Quentin Dominique Thomas, Mohamed Chaabouni, Anas Al herk, Cesar Lefevbre, Sarah Cavaillon, Léa Sinoquet, Stéphane Pouderoux, Marie Viala, Lise Roca, Xavier Quantin

**Affiliations:** 1Department of Medical Oncology, Montpellier Cancer Institute (ICM), University of Montpellier (UM), 34090 Montpellier, France; mohamed.chaabouni@icm.unicancer.fr (M.C.); sarah.cavaillon@icm.unicancer.fr (S.C.); lea.sinoquet@icm.unicancer.fr (L.S.); stephane.pouderoux@icm.unicancer.fr (S.P.); marie.viala@icm.unicancer.fr (M.V.); xavier.quantin@icm.unicancer.fr (X.Q.); 2Oncogenic Pathways in Lung Cancer, Montpellier Cancer Research Institute (IRCM) INSERM U1194, University of Montpellier (UM), 34090 Montpellier, France; 3Biometrics Unit ICM, Montpellier Cancer Institute, University of Montpellier (UM), 34090 Montpellier, France; anas.alherk@icm.unicancer.fr (A.A.h.); lise.roca@icm.unicancer.fr (L.R.); 4Pharmacy Department, Montpellier Cancer Institute (ICM), 34090 Montpellier, France; cesar.lefevbre@icm.unicancer.fr

**Keywords:** non-small-cell lung cancer, immunotherapy, geriatric oncology, weekly chemotherapy, prognosis

## Abstract

**Simple Summary:**

The use of immune checkpoint blockers targeting PD-1 is a standard therapy in combination with chemotherapy for patients with non-small-cell lung cancer (NSCLC) in the first metastatic line. The efficacy and tolerability of this combination is currently unknown for frail patients (i.e., elderly patients and/or poor Eastern Cooperative Oncology Group Performance Status). We treated patients with a combination of carboplatin (AUC 5 mg/mL/min; Q4W), weekly paclitaxel (90 mg/m^2^ on days 1, 8, and 15), and pembrolizumab (200 mg Q4W). We observed an overall response rate of 55.9%. Median real-world progression-free survival was 10.6 months (95% CI [6.0, NA]). Median overall survival (OS) was not reached, with 12- and 18-month OS rates of 75.6% and 61.4%. This chemoimmunotherapy combination demonstrates promising efficacy in frail patients with metastatic NSCLC. The safety profile of this combination was comparable to those of the standard of care in the first metastatic line. Prospective clinical trials are warranted to confirm these real-word results.

**Abstract:**

Introduction: Immune checkpoint blockers have revolutionized the first-line treatment of advanced non-small-cell lung cancer (NSCLC). Pembrolizumab, an anti-PD-1 monoclonal antibody, is a standard therapy either alone or in combination with chemotherapy (chemo-IO). The current study explores the efficacy and safety of pembrolizumab with carboplatin and weekly paclitaxel in a cohort of frail patients. Methods: A monocentric retrospective study was conducted between 22 September 2020 and 19 January 2023 regarding patients with stage IV NSCLC treated with chemo-IO combination: carboplatin (AUC 5 mg/mL/min; Q4W), weekly paclitaxel (90 mg/m^2^ on days 1, 8, and 15), and pembrolizumab (200 mg Q4W). The primary objective was real-world progression-free survival (rwPFS). Secondary objectives were overall survival (OS), toxicity profile, and outcomes based on histological subtype. Results: A total of 34 patients (20 squamous and 14 non-squamous NSCLC) benefited from the chemo-IO regimen for frail patients; 41.9% had an ECOG-PS = 2. The median age was 75.5 years. We observed an overall response rate (ORR) of 55.9%. Notably, squamous NSCLC exhibited a significantly higher ORR (80%) than non-squamous NSCLC (21.4%); *p* = 0.001. The median rw-PFS was 10.6 months (95% CI [6.0, NA]), with 6- and 12-month rw-PFS rates of 69% and 45.8%, respectively. The median OS was not reached, with 12- and 18-month OS rates of 75.6% and 61.4%, respectively. The median number of maintenance cycles of pembrolizumab was 5 (0; 27). Nine patients (26.5%) experienced a toxicity related to chemotherapy leading to a reduction of the dose administered and, in five patients (14.7%), to the permanent discontinuation of chemotherapy. Six patients (17.6%) had an immune-related adverse event leading to the discontinuation of immunotherapy. Discussion: Pembrolizumab plus carboplatin and weekly paclitaxel demonstrates promising efficacy and safety in frail patients with metastatic NSCLC, especially for ORR in sq-NSCLC. Prospective studies focusing on frail populations are warranted in order to validate these findings and optimize therapeutic strategies in the first-line setting.

## 1. Introduction

Non-small-cell lung cancer (NSCLC) remains the leading cause of cancer-related deaths worldwide [1]. With the approval of immune checkpoint blockers (ICBs), the last decade has witnessed a transformative shift in the therapeutic approach for treatment-naïve patients with advanced NSCLC lacking oncogenic drivers [2]. In the first line of metastatic treatment, the standard of care (SoC) involves pembrolizumab, an anti-programmed death protein 1 (PD-1) monoclonal antibody, administered either alone (mono-IO) or in combination with chemotherapy (chemo-IO) [3,4,5,6]. The KEYNOTE-407 study, focusing on squamous NSCLC (sq-NSCLC), demonstrated that patients receiving pembrolizumab in combination with chemotherapy exhibited positive outcomes. For the initial four cycles, carboplatin (AUC 6 mg/mL/min) was administered on day 1, along with either paclitaxel (200 mg/m^2^) on day 1 or nab-paclitaxel (100 mg/m^2^) on days 1, 8, and 15. Following induction treatment, patients received either pembrolizumab every 3 weeks (Q3W) or a placebo for up to 35 cycles [4]. In the KEYNOTE-189 study, assessing pembrolizumab plus chemotherapy for non-squamous NSCLC (nsq-NSCLC), patients received cisplatin (75 mg/m^2^) or carboplatin (AUC 5 mg/mL/min) with pemetrexed (500 mg/m^2^) Q3W intravenously. After the induction treatment, patients received pemetrexed (500 mg/m^2^) Q3W with or without pembrolizumab for up to 35 cycles [5]. These phase III trials included patients with an Eastern Cooperative Oncology Group Performance Status (ECOG-PS) score of 0 or 1. While there was no upper age limit for inclusion, the proportion of patients ≥ 75 years old in these studies remained below 10%.

Before the era of ICBs in advanced NSCLC, carboplatin and weekly paclitaxel doublet chemotherapy demonstrated overall survival (OS) benefits compared to monotherapy (vinorelbine or gemcitabine) in the first metastatic line for elderly patients (≥70 years old). In the IFCT-0501 trial, the median OS was 10.3 months for doublet chemotherapy and 6.2 months for monotherapy (hazard ratio = 0.64; 95% CI 0.52–0.78; *p* < 0.0001) [6]. Patients assigned to doublet chemotherapy received intravenous carboplatin AUC 6 on day 1 and 90 mg/m^2^ paclitaxel on days 1, 8, and 15. Cycles were repeated every 4 weeks (3 weeks of treatment plus 1 week without) [7]. The efficacy and tolerability of immunotherapy in combination with a chemotherapy doublet are currently unknown for frail patients with metastatic NSCLC. Here, we report the results of a monocentric retrospective study evaluating these criteria in a population of NSCLC patients treated in the first metastatic line with monthly carboplatin and weekly paclitaxel in combination with pembrolizumab.

## 2. Methods

### 2.1. Study Design and Participant Selection

We established an institutional database to collect information from patients who received treatment at the Montpellier Cancer Institute (ICM) in France, one of the eighteen comprehensive cancer centers in the country. Specifically, we focused on frail patients who underwent treatment involving a combination of pembrolizumab with carboplatin and weekly paclitaxel. Frail patients have been defined as patients aged ≥ 70 years and/or with poor ECOG-PS = 2. This database received approval from the ICM institutional review board. Patients were included in the study according to the following criteria:
Histologically proven NSCLC. Cytological evidence was authorized.ECOG-PS ≤ 2.Stage IV NSCLC according to TNM 8^th^ classification, UICC 2015.Absence of systemic anticancer therapy given as first-line treatment for advanced or metastatic disease.

The exclusion criteria were as follows:
Small cell lung cancer (SCLC) or tumor with mixed histology, including a small cell component.Known EGFR activating mutation.Known ALK or ROS-1 gene rearrangement assessed by immunohistochemistry, FISH, or NGS sequencing.Polyneuropathy of CTCAE v5.0 grade ≥ 2.

Our study population comprised patients who were treated for advanced NSCLC using the following regimen: carboplatin (AUC 5 mg/mL/min; Q4W) and paclitaxel (90 mg/m^2^ on days 1, 8, and 15; a 3-week treatment cycle with 1 week off) for the first four cycles, in combination with pembrolizumab (200 mg Q4W). After the induction treatment, patients received pembrolizumab at a dose of 200 mg Q3W, which continued until disease progression or discontinuation due to toxicity.

### 2.2. Objectives of the Study

The primary objective of this study was to assess real-world progression-free survival (rwPFS) in patients treated with the chemo-IO combination. RwPFS was defined as the time elapsed from the initial treatment administration to the first documented instance of disease progression or death from any cause (whichever occurred first), with the last date of follow-up considered as the censoring point. For patients who discontinued treatment due to severe side effects, the same definition has been used.

The secondary objectives of the study included the following:Evaluating overall survival (OS), which was defined as the time from the initiation of treatment to death from any cause, with the last date of follow-up serving as the censoring point.Evaluating rwPFS and OS according to the histological type of NSCLC (sq-NSCLC and nsq-NSCLC)Identifying demographic characteristics of the treated patients.Assessing the toxicity profile of this combination therapy within predefined subgroups.

### 2.3. Data Collection and Statistical Analysis

We recorded a comprehensive set of data, including patient demographics (age, sex, Eastern Cooperative Oncology Group Performance Status (ECOG-PS), smoking status, history of other cancers, renal function, autoimmune antibodies), tumor characteristics (histology, presence of brain metastases, PD-L1 tumor proportion score), and treatment history (prior therapies for nonmetastatic disease, subsequent lines of treatment).

Response to treatment was evaluated per investigator according to the Response Evaluation Criteria in Solid Tumors (RECIST), version 1.1, which categorized patients into the following groups: complete response (CR), partial response (PR), stable disease (SD), and progression disease (PD) [8]. We also documented toxicities associated with chemotherapy and immunotherapy, classifying them into four predefined groups: toxic events leading to death, toxic events resulting in discontinuation of immunotherapy, =toxic events necessitating dose reduction of chemotherapy, and toxic events leading to permanent discontinuation of chemotherapy. Immune-related adverse events were classified according to the Common Terminology Criteria for Adverse Events (CTCAE) v.5.0.

Objective response rate (ORR) was defined as the percentage of patients with CR and PR. Disease control rate (DCR) was defined as the percentage of patients with CR, PR, and SD as the best response at the database cutoff date.

Continuous variables were described by the number of observations (N), the median, minimum, maximum, mean, and standard deviation. Student’s t-test or the Wilcoxon test were used to compare the distribution of continuous variables. Categorical variables were described by the number of observations (N) and the frequency (%) of each modality. Missing categories were counted. Percentages were calculated in the overall population excluding missing data. The Chi-square test was used to compare proportions (or the Fisher’s exact test if the expected frequencies were <5). Median follow-up was estimated using the reverse Kaplan–Meier method for OS and PFS. The Kaplan–Meier method was used to estimate survival rates and median survival times and their associated 95% confidence interval (95% CI). The survival distribution of both treatment arms was compared using the log-rank test. Data quality validation, data preparation, and survival analysis were conducted using R 4.1.3. Descriptive analysis was conducted using SAS 9.4 adclin programs. All statistical tests were two-sided, and the significance threshold was set at 5% (i.e., *p* < 0.05).

## 3. Results

### 3.1. Population Demographics

Thirty-four patients with stage IV NSCLC were treated with pembrolizumab in combination with carboplatin and weekly paclitaxel between 22 September 2020 and 19 January 2023 in our institution (Figure 1). At inclusion, according to the demographic characteristics of patients (Table 1), 29 patients (85.3%) were men, the median age was 75.5 years, and 79.4% were 70 years old or older. As many as 41.9% had an ECOG-PS = 2. A total of 25 patients (73.5%) were de novo metastatic. Most of the patients were PD-L1-negative (53.1%). Four patients (12.1%) had cerebral metastasis at diagnosis. Regarding histological subtypes, 20 patients (58.8%) included had sq-NSCLC, and 14 patients (41.2%) had nsq-NSCLC. Of note, nsq-NSCLC patients tended to be frailer in comparison with sq-NSCLC patients with regard to age ≥ 70 years old (85.7% vs. 75.0%), ECOG-PS = 2 (53.8% vs. 33.3%), and chronic kidney failure (35.7% vs. 10.0%).

### 3.2. Treatment Administration and Toxicites

The standard administered regimen was carboplatin (AUC 5 mg/mL/min; Q4W) and paclitaxel (90 mg/m^2^ on days 1, 8, and 15; a 3-week treatment cycle with 1 week off) for the first four cycles in combination with pembrolizumab (200 mg Q4W). Due to their general status and/or comorbidities, 10 patients (29.4%) had a baseline adaptation of the chemotherapy regimen: carboplatin (AUC 4 mg/mL/min) for 3 patients; paclitaxel (80 mg/m^2^) for 4 patients; and both therapeutic adaptations for 3 patients. Most patients 22/34 (64.7%) presented a toxic event related to chemotherapy and/or immunotherapy. Nine patients (26.5%) experienced a toxicity related to chemotherapy leading to a reduction of the dose administered and, in five patients (14.7%), to the permanent discontinuation of chemotherapy. No toxic events related to chemotherapy leading to death were reported. Six patients (17.6%) had an immune-related adverse event (irAEs) leading to the discontinuation of immunotherapy, among which one patient died due to hepatitis induced by immunotherapy. There was no significant difference in the safety profile of treatment according to histological subtype (Table A1).

### 3.3. Survival Outcomes: Control Rate, Progression-Free Survival, and Overall Survival

In the cohort of 34 patients, the objective response rate was 55.9%, with 19 patients presenting partial response (no CR were observed). Disease control rate was obtained for 30/34 patients (88.2%) with 32.4% of stable disease. According to the histological subtype, 16 sq-NSCLC (80%) versus 3 nsq-NSCLC (21.4%) had an ORR (*p* = 0.001), and 19 sq-NSCLC (95.0%) versus 11 nsq-NSCLC (78.6%) had a DCR (*p* = 0.28). Twenty-six patients (76.5%) achieved the induction with four cycles of chemo-immunotherapy. The median number of maintenance cycles of pembrolizumab was five, independently of the histological subtype. Regarding patients presenting progression after the first metastatic line with chemoimmunotherapy, 15/19 (78.9%) benefited from at least one subsequent line of treatment. Of note, no patients with nsq-NSCLC benefited from more than one subsequent line of treatment after progression (Table 2).

With a median follow-up time of 9.5 months [8.2, 12.6], the median rw-PFS was 10.6 months (95% CI [6.0, NA]). The 6-months rw-PFS was 69% (95% CI [52.9%, 85.1%]); and the 12-months rw-PFS was 45.8% (95% CI [27.0%, 64.6%]) (Figure 2). By stratifying according to the histological subtype, the 6-months rw-PFS was 71.4% (95% CI [47.8%, 95.1%]); and 12-months rw-PFS was 51.0% (95% CI [21.7%, 80.3%]) for nsq-NSCLC. The 6-months rw-PFS was 67.3% (95%CI [45.7%, 88.9%]); and 12-months rw-PFS was 43.6% (95% CI [20.2%, 67.0%]) for sq-NSCLC. There were no differences regarding median rw-PFS according to histological subtype (sq vs. nsq-NSCLC); *p* = 0.77 (Figure A1).

At the time of data analysis, the median OS had not been reached (95% CI [16.8, NA]). The 12-months OS was 75.6% (95% CI [59.3%, 91.9%]); and the 18-months OS was 61.4% (95% CI [38.8%, 84.1%]). By stratifying according to the histological subtype, the 12-months OS was 68.8% (95% CI [35.1%, 100%]); and the 18-months OS was 68.8% (95% CI [35.1%, 100%]) for nsq-NSCLC. The 12-months OS was 77.6% (95% CI [58.2%, 97.0%]); and the18-months OS was 55.9% (95% CI [25.4%, 86.4%]) for sq-NSCLC (Figure 2).

## 4. Discussion

The combination of ICBs with platinum-based doublet chemotherapy represents the SoC for metastatic NSCLC in first-line treatment. In this report, we present findings from a pivotal study involving 34 patients (20 with squamous-cell NSCLC and 14 with non-squamous NSCLC) treated with pembrolizumab in combination with carboplatin and weekly paclitaxel. Across the entire cohort, the ORR was 55.9%, and the median rw-PFS was 10.6 months (95% CI [6.0, NA]), with a median OS that was not reached (95% CI [16.8%, NA]) over a median follow-up of 9.5 months [8.2, 12.6]. These results provide reassurance regarding the feasibility and efficacy of this combination in the first-line metastatic treatment of NSCLC and align with findings from randomized phase III clinical trials reporting median PFS of 9 months (95% CI [8.1, 10.4]) for nsq-NSCLC and 8 months (95% CI [6.3, 8.5]) for sq-NSCLC [9,10]. Notably, our cohort exhibited greater frailty, with a median age of 75.5 years and a 41.9% Eastern Cooperative Oncology Group Performance Status (ECOG-PS) = 2—more specifically for nsq-NSCLC, for which 85.7% were ≥70 years old and 53.8% were ECOG-PS = 2. Of note, the frequency of irAEs leading to discontinuation of immunotherapy in our cohort (17.6%) is in line with the KEYNOTE-189 study (20.2%) for nsq-NSCLC [9] and the KEYNOTE-407 study (17.3%) for sq-NSCLC [10].

Potential contributors to our results include the implementation of supportive care for all patients and oncogeriatric evaluation for 7/27 (25.9%) of elderly patients (i.e., age ≥ 70 years old), as well as the immunostimulatory effects of taxanes. Weekly paclitaxel administration at the initiation of ICB treatment may offer advantages, such as enhanced dose intensity and potential immune priming, contributing to increased sensitivity to immunotherapy [11,12]. Our cohort demonstrated a significant difference in ORR, favoring sq-NSCLC (80%) over nsq-NSCLC (21.4%). Similar findings were observed in randomized phase III clinical trials, where sq-NSCLC exhibited a higher ORR (57.9%; 95% CI, 51.9 to 63.8) compared to nsq-NSCLC (47.6%; 95% CI, 42.6 to 52.5) [4,5]. Possible explanations for these differences in our cohort include the greater frailty of nsq-NSCLC patients, who may be unable to receive the standard of care with pemetrexed chemotherapy, or the limited sample size of our cohort.

Our results align with the recent phase II “Frail-Immune” trial, which evaluated the efficacy and safety of durvalumab combined with weekly paclitaxel and carboplatin in first-line treatment for recurrent/metastatic squamous cell carcinoma of the head and neck (R/M SCCHN). The trial demonstrated a median overall survival of 18.0 months (95% CI [11.9-NR]) and a 24-month overall survival rate of 45% (32–57%) [13]. These positive outcomes suggest that weekly administration of chemotherapy in combination with ICBs can be effective. Despite the non-randomized nature of the trial, the results surpassed the current SoC for first-line metastatic treatment of R/M SCCHN with pembrolizumab, either alone or combined with platinum-5FU, which achieved a median OS of 13 months [14].

Real-life or academic studies are crucial for understanding therapeutic strategies in elderly and frail patients, who are often underrepresented in randomized trials. For instance, the ESME Advanced or Metastatic Lung Cancer (AMLC) data platform revealed that only 6.7% of patients aged 70 years or older were enrolled in clinical trials for first metastatic line setting [15]. Addressing the optimal therapeutic strategy for frail patients with high PD-L1 expression (i.e., ≥50%) is a critical consideration. Currently, patients with good performance status could be treated with either mono-IO or chemo-IO [16,17]. Emerging evidence suggests that chemo-IO may yield better survival outcomes in terms of ORR, PFS, and a potential trend for improved OS [18,19]. Clinical limitations for proposing chemo-IO in patients with high PD-L1 expression may include those with altered performance status (ECOG PS-2 or 3) or elderly patients, for whom mono-IO is more easily recommended [16]. Several randomized trials have demonstrated that weekly administration of paclitaxel in combination with carboplatin can reduce the hematotoxicity and neurotoxicity compared to triweekly paclitaxel administration [20,21,22,23]. Administering a less toxic platinum-doublet chemotherapy may theoretically facilitate the proposal of chemoimmunotherapy, even for elderly patients or those with less favorable general conditions (i.e., ECOG-PS 2). The ongoing IFCT-1805 ELDERLY trial (NCT03977194) is recruiting elderly patients (70–89 years old with ECOG-PS 0 or 1) in the first metastatic line treatment for NSCLC (either squamous or non-squamous). Patients will be randomized to receive carboplatin (AUC 5 mg/mL/min; Q4W) and paclitaxel (90 mg/m^2^; on days 1, 8, and 15 of each cycle) with or without atezolizumab 1200 mg Q3W. The results of this trial will provide insights into the optimal therapeutic strategy for elderly patients with first-line metastatic NSCLC.

Our study has some limitations that need to be highlighted. The retrospective design and limited number of included patients reduce the statistical power of our study, potentially explaining the wide variation in ORR between squamous and non-squamous NSCLC. Additionally, we chose to administer pembrolizumab at the standard flat-dose (200 mg) Q4W, deviating from the SoC, which prescribes pembrolizumab at 200 mg Q3W or 400 mg Q6W [16]. This decision aimed to enhance the quality of life for patients by minimizing the number of chemo-IO administrations (i.e., no administration on day 21). Although there are no pharmacokinetic (PK) data evaluating pembrolizumab administration Q4W, PK exposure metrics for 400 mg of pembrolizumab Q6W after the first treatment cycle show a concentration variation of less than 10 μg/mL between day 20 and day 30 after ICB administration [24]. Moreover, pharmacokinetics and pharmacodynamics studies of patients treated by pembrolizumab have been developed according to 3 + 3 dose escalation design [25]. This design of phase I clinical trials has some drawbacks. The result from the current dose is used to determine the dose of the next cohort of patients, and information on other doses is ignored. Furthermore, this method leaves doubts regarding the optimal recommended phase II dose (RP2D) when no maximum tolerated dose (MTD) is observed [26,27,28]. Indeed, no dose-limiting toxicities were observed in the phase I study of pembrolizumab [25]. From this perspective, a recent phase III superiority trial of nivolumab (an anti-PD-1 monoclonal antibody) administered at 20 mg flat dose once every 3 weeks in combination with triple metronomic chemotherapy has resulted in a significant improvement of OS for metastatic R/M SCCHN versus triple metronomic chemotherapy alone [29]. The effectiveness of this low-dose immunotherapy, whereas the standard dose of nivolumab for R/M SCCHN is 3 mg/kg once every 2 weeks in the second metastatic line, reinforces the possibility of overexposure of patients treated by ICBs [30]. These results suggest a low impact of administering pembrolizumab 200 mg Q4W. Moreover, a randomized non-inferiority phase III clinical trial (NCT05692999) is currently assessing a new mode of immunotherapy administration, based on an increased interval time between two infusions as maintenance treatment, compared with conventional administration for metastatic nsq-NSCLC in the first metastatic line. The experimental arm involves administering pembrolizumab 200 mg Q6W plus, in the absence of contraindication, pemetrexed 500 mg/m^2^ Q3W.

In our institution, we actually treat the majority of our patients (64.2%) with sq-NSCLC in the first metastastic line with this combination of pembrolizumab with carboplatin and weekly paclitaxel (Figure 1). These data will have to be validated in prospective randomized trials.

## 5. Conclusions

Pembrolizumab in combination with carboplatin and weekly paclitaxel shows promising results in terms of its efficacy and safety profiles for frail patients with metastatic NSCLC. Prospective studies evaluating this chemo-IO regimen in the first metastatic line for frail patients with NSCLC are warranted.

## Figures and Tables

**Figure 1 cancers-16-00992-f001:**
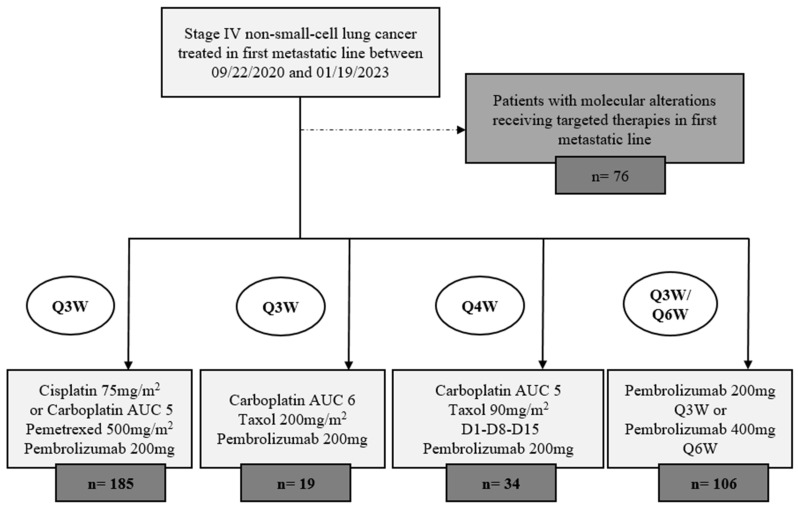
Flowchart.

**Figure 2 cancers-16-00992-f002:**
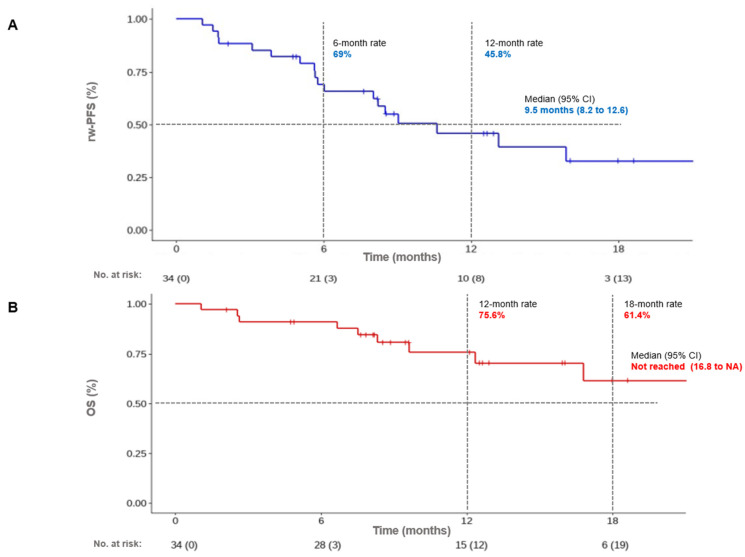
Progression-free survival (**A**) and overall survival (**B**) in the entire cohort.

**Table 1 cancers-16-00992-t001:** Demographic and disease characteristics of the patients at baseline.

	Total Cohort (*n* = 34)	Sq-NSCLC (*n* = 20)	Nsq-NSCLC (*n* = 14)	*p*-Value
**Age at date of first treatment cycle (yr)**				
Median (range)	75.5 (61; 81)	74.5 (61; 81)	78.0 (67; 81)	0.09
≥70 yr—no. (%)	27 (79.4)	15 (75.0)	12 (85.7)	0.67
**Male sex—no. (%)**	29 (85.3)	17 (85.0)	12 (85.7)	1
**Smoking statuts—no. (%)**				1
Current or former	34 (100)	20 (100)	14 (100)	
Never	0 (0)	0 (0)	0 (0)	
**ECOG performance status score—no. (%)**	(*n* = 31)	(*n* = 18)	(*n* = 13)	0.42
0	7 (22.6)	4 (22.2)	3 (23.1)	
1	11 (35.5)	8 (44.4)	3 (23.1)	
2	13 (41.9)	6 (33.3)	7 (53.8)	
**Renal function GFR (mL/min/m^2^)—no. (%)**				0.07
≥60	27 (79.4)	18 (90.0)	9 (64.3)	
45–60	4 (11.8)	2 (10.0)	2 (14.3)	
30–45	3 (8.8)	0 (0)	3 (21.4)	
**Brain metastases—no. (%)**	(*n* = 33)	(*n* = 20)	(*n* = 13)	1
	4 (12.1)	3 (15.0)	1 (7.7)	
**PD-L1 tumor proportion score—no. (%)**	(*n* = 32)	(*n* = 18)	(*n* = 14)	0.40
<1%	17 (53.1)	8 (44.4)	9 (64.3)	
1–49%	13 (40.6)	8 (44.4)	5 (35.7)	
≥50%	2 (6.3)	2 (11.1)	0 (0)	
**Previous therapy for non-metastatic disease**				0.26
De novo metastatic	25 (73.5)	16 (80.0)	9 (64.3)	
Surgery	3 (8.8)	1 (5.0)	2 (14.3)	
Stereotactic radiotherapy	2 (5.9)	0 (0.0)	2 (14.3)	
Radiochemotherapy	4 (11.8)	3 (15.0)	1 (7.1)	

**Table 2 cancers-16-00992-t002:** Treatment regimens and outcomes.

	Total Cohort (*n* = 34)	Sq-NSCLC (*n* = 20)	Nsq-NSCLC (*n* = 14)	*p*-Value
**Chemotherapy regimen at baseline—no. (%)**				*p* = 0.86
Carboplatin AUC 5	28 (82.3)	17 (85.0)	11 (78.6)	
Carboplatin AUC 4	6 (17.7)	3 (15.0)	3 (21.4)	
Taxol 90 mg/m^2^	27 (79.4)	15 (75.0)	12 (85.7)	
Taxol 80 mg/m^2^	7 (20.6)	5 (25.0)	2 (14.3)	
**Best response to treatment—no. (%)**				
Complete response (CR)	0	0	0	
Partial response (PR)	19 (55.9)	16 (80.0)	3 (21.4)	
Stable disease (SD)	11 (32.4)	3 (15.0)	8 (57.2)	
Progression disease (PD)	4 (11.7)	1 (5.0)	3 (21.4)	
Objective response rate (CR + PR)	19 (55.9)	16 (80.0)	3 (21.4)	*p* = 0.001
Disease control rate (CR + PR + SD)	30 (88.2)	19 (95.0)	11 (78.6)	*p* = 0.28
**Immunotherapy: Number of maintenance cycles**				*p* = 0.90
Median (range)	5.0 (0; 27)	5.0 (0; 27)	5.0 (0; 21)	
None	8 (23.5)	4 (20.0)	4 (28.6)	
0–5	9 (26.5)	6 (30.0)	3 (21.4)	
>5	17 (50.0)	10 (50.0)	7 (50.0)	
**Number of subsequent lines of treatment—no. (%)**				*p* = 0.67
0	19 (55.9)	10 (50.0)	9 (64.3)	
1	12 (35.3)	7 (35.0)	5 (35.7)	
2	2 (5.9)	2 (10.0)	0 (0.0)	
3	1 (2.9)	1 (5.0)	0 (0.0)	

## Data Availability

The data presented in this study are available on request from the corresponding author (quentin.thomas@icm.unicancer.fr).

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
