# Peer review of "Exploring the Efficacy of Pembrolizumab in Combination with Carboplatin and Weekly Paclitaxel for Frail Patients with Advanced Non-Small-Cell Lung Cancer: A Key Investigative Study"

_cancers, 2024, doi:10.3390/cancers16050992_

Round 1
Reviewer 1 Report
Comments and Suggestions for Authors
I had the privilege of reviewing your manuscript titled "Exploring the Efficacy of Pembrolizumab in Combination with Carboplatin and Weekly Paclitaxel for Frail Patients with Advanced Non-small Cell Lung Cancer: A Key Investigative Study." First, allow me to commend you on the well-written introduction and the choice of focus for your research. Investigating treatment options for frail patients with advanced non-small cell lung cancer is not only innovative but also immensely crucial for tailoring cancer care to this vulnerable population's unique needs.
Strengths:
The manuscript is well-structured with a clear narrative that guides the reader through the study's rationale, methodology, and findings.
The focus on frail patients, a group often underrepresented in clinical research, is particularly commendable. This study contributes valuable insights into personalized treatment strategies for these individuals.
Comments:
Definition of Frail Patients: Could you please provide a more detailed definition of 'frail patients' as used in your study? A clear understanding of this term is essential for replicating your study and for readers to accurately interpret your findings.
Accounting for Discontinued Treatments Due to Side Effects: It would be beneficial to include information on how patients who discontinued treatment due to side effects were accounted for in your analysis. This detail will enhance the transparency and reliability of your study's outcomes.
Misalignment in Tables 1 and 2: I have observed that there are misaligned cells within Table 1 and Table 2, which could potentially lead to confusion or misinterpretation of the data presented. Could you kindly review and correct the alignment of cells in these tables to ensure the data is accurately and clearly presented?
Author Response
First of all, we would like to thank reviewer 1 for the time dedicated to proofreading our manuscript and for the enthusiasm shared about the topic research of our research. Here is a point-by-point answer to the comments of the reviewer :
- Definition of Frail Patients: indeed a precise definition of frail patients is needed in our manuscript to help the interpretation of the results and enable them to be reproduced. Here is the definition add on our manuscript: “we focused on frail patients who underwent treatment involving a combination of pembrolizumab with carboplatin and weekly paclitaxel. Frail patients have been defined as patients aged ≥70 years and/or with poor ECOG-PS = 2” (lines 97-100).
-Accounting for discontinued treatments due to side effects: we would like to thank the reviewer for vigilance on this point. We have considered patients who had discontinuated the treatment due to severe side effects related to immunotheray as others for the definition of PFS and OS. The statistical point date considered was the date of observed progression disease or date of death from any cause. Additional information has been added to the methods:
“The primary objective of this study was to assess real-world progression-free survival (rwPFS) in patients treated with the chemo-IO combination. RwPFS was defined as the time elapsed from the initial treatment administration to the first documented instance of disease progression or death from any cause, whichever occurred first, with the last date of follow-up considered as the censoring point. For patients who discontinued treatment due to severe side effects the same definition has been used.“ (lines 121-126)
- Misalignment in Tables 1 and 2: The correction was made in the main file (pages 6-7)

Reviewer 2 Report
Comments and Suggestions for Authors
The authors intended to explore the Efficacy of Pembrolizumab in Combination with Carboplatin and Weekly Paclitaxel for Frail Patients with Advanced Non-small Cell Lung Cancer. The topic was really interesting, and the ICB dramatically increased the clinical effects of NSCLC and other cancers in recent years. And the authors performed a clinical trial and proved the effects of ICB with chemotherapy. The median rw- PFS was 10.6 months (95% CI [6.0, NA]), with 6- and 12-month rw-PFS rates of 69% and 45.8%, respectively. The results were acceptable,
However, there were some weak points:
1. Some paper already indicted that the efficacy of Pembrolizumab in Combination with chemotherapy, such as: J Thorac Oncol. 2020 Oct;15(10):1657-1669, J Clin Oncol. 2023 Apr 10;41(11):1999-2006. Lung Cancer. 2021 May;155:175-182. The authors should explain the Innovation of this manuscript.
2. The quantity of patients were only 34, could they add more patients to the research?
3. Please mention the methods of statistics of this manuscript more clearly.
Comments on the Quality of English LanguagePlease opimize the discussion.
Author Response
First of all, we would like to thank reviewer 2 for the time dedicated to proofreading our manuscript and for the enthusiasm shared about the topic research of our research. Here is a point-by-point answer to the comments of the reviewer :
-Regarding the articles reffered to by the reviewer (J Thorac Oncol. 2020 Oct;15(10):1657-1669, J Clin Oncol. 2023 Apr 10;41(11):1999-2006. Lung Cancer. 2021 May;155:175-182.), these articles focus on combination of chemo-immunotherapy regimen with administration of chemotherapy every 3 weeks. The original design of our study is the weekly administration of chemotherapy (i.e. weekly paclitaxel) in combination with immune checkpoint inhibitors. To the best of our knowledge this chemotherapy regimen adapted for frail patients has never been evaluated in combination with immunotherapy for patient treated for advanced NSCLC in first metastatic line. The originality of this study has been explained in the introduction of our manuscript. We remain at the reviewer’s disposal if he considers that we need to go into more detail on this issue.
-We totally agree with the reviewer that the number of patients included in our study represents a major limitation of our manuscript. Unfortunately iy would take several years for us to have a large cohort as demonstrated by a total number of 34 inclusion between 09/22/2020 and 01/19/2023. The monocentric design of our study implies the small sample size. We hope that our results are convincing enought to be reproduced in a prospective multicentric trial to obtain more consistent results from a larger cohort.
-Concerning the methods of statistics of the manuscript : indeed our statistical methodology needed to be improved from the first manuscript. We have added a paragraph to precise the methodology used : ”Continuous variables were described by the number of observations (N), the median, minimum, maximum, mean and standard deviation. The Student t-test or Wilcoxon test were used to compare the distribution of continuous variables. Categorical variables were described by the number of observations (N) and the frequency (%) of each modality. Missing categories were counted. Percentages were calculated in the overall population excluding missing data. The Chi-square test was used to compare proportions (or the Fisher's exact test if the expected frequencies were <5). Median follow-up was estimated using the reverse Kaplan-Meier method for OS and PFS. The Kaplan-Meier method was used to estimate survival rates and median survival times and their associated 95% confidence interval (95% CI). The survival distribution of both treatment arms was compared using the Log-rank test. Data quality validation, data preparation, and survival analysis were conducted using R 4.1.3. Descriptive analysis was conducted using SAS 9.4 adclin programs. All statistical tests were two-sided and the significance threshold was set at 5% (i.e. p < 0.05).” (lines 155-168)
-An optimization of the discussion has been made focusing on the pharmacokinetics of immune checkpoint inhibitors : ”Moreover, pharmacokinetics and pharmacodynamics studies of patients treated by pembrolizumab have been developed according to 3+3 dose escalation design [25]. This design of phase I clinical trials have some drawbacks. The result from the current dose is used for determining the dose of next cohort of patients and information on other doses is ignored. Furthermore, this method leaves doubts regarding the optimal recommended phase II dose (RP2D) when no maximum tolerated dose (MTD) is observed [26-28]. Indeed, no dose-limiting toxicities were observed in the phase I study of pembrolizumab [25]. In this perspective, a recent phase III superiority trial of nivolumab (an anti-PD-1 monoclonal antibody) administred at 20mg flat dose once every 3 weeks in combination with triple metronomic chemotherapy has resulted in a significant improvement of OS for metastatic R/M SCCHN versus triple metronomic chemotherapy alone [29]. The effectiveness of this low-dose immunotherapy whereas the standard dose of nivolumab for R/M SCCHN is 3mg/kg once every 2 weeks in second metastatic line reinforces the possibility of overexposure of patients treated by ICBs [30].” (lines 310-324)

Round 2
Reviewer 2 Report
Comments and Suggestions for Authors
Thanks for submit the revised vision. Minor English revise is required.
Comments on the Quality of English Language
Thanks for submit the revised vision. Minor English revise is required.